# On the Nature and Origin of Atmospheric Annual and Semi-Annual Oscillations

**Vincent Courtillot [1], Jean-Louis Le Mouël [1], Fernando Lopes [1,\* and Dominique Gibert [2]**

1   Institut de Physique du globe de Paris, Université de Paris, 75005 Paris, France
2   LGL-TPE-Laboratoire de Géologie de Lyon-Terre, Planètes, Environnement, 69622 Lyon, France
\*   Correspondence: lopesf@ipgp.fr

**Abstract:** This paper proposes a joint analysis of variations of global sea-level pressure (SLP) and of Earth's rotation (RP), expressed as the coordinates of the rotation pole ($m_1$, $m_2$) and length of day (lod). We retain iterative singular spectrum analysis (iSSA) as the main tool to extract the trend, periods, and quasi periods in the data time series. SLP components are a weak trend, seven quasi-periodic or periodic components ($\sim$130, 90, 50, 22, 15, 4, 1.8 years), an annual cycle, and its first three harmonics. These periods are characteristic of the space-time evolution of the Earth's rotation axis and are present in many characteristic features of solar and terrestrial physics. The amplitudes of the annual SLP component and its three first harmonics decrease from 93 hPa for the annual to 21 hPa for the third harmonic. In contrast, the components with pseudo-periods longer than a year range between 0.2 and 0.5 hPa. We focus mainly on the annual and, to a lesser extent, the semi-annual components. The annual RP and SLP components have a phase lag of 152 days (half the Euler period). Maps of the first three components of SLP (that together comprise 85% of the data variance) reveal interesting symmetries. The trend is very stable and forms a triskeles structure that can be modeled as Taylor–Couette flow of mode 3. The annual component is characterized by a large negative anomaly extending over Eurasia in the NH summer (and the opposite in the NH winter) and three large positive anomalies over Australia and the southern tips of South America and South Africa in the SH spring (and the opposite in the SH autumn), forming a triskeles. The semi-annual component is characterized by three positive anomalies (an irregular triskeles) in the NH spring and autumn (and the opposite in the NH summer and winter), and in the SH spring and autumn by a strong stable pattern consisting of three large negative anomalies forming a clear triskeles within the 40–60° annulus formed by the southern oceans. A large positive anomaly centered over Antarctica, with its maximum displaced toward Australia, and a smaller one centered over Southern Africa, complement the pattern. Analysis of iSSA components of global sea level pressure shows a rather simple spatial distribution with the principal forcing factor being changes in parameters of the Earth's rotation pole and velocity. The flow can probably best be modeled as a set of coaxial cylinders arranged in groups of three (triskeles) or four and controlled by Earth topography and continent/ocean boundaries. Flow patterns suggested by maps of the three main iSSA components of SLP (trend, annual, and semi-annual) are suggestive of Taylor–Couette flow. The envelopes of the annual components of SLP and RP are offset by four decades, and there are indications that causality is present in that changes in Earth rotation axis lead force pressure variations.

**Keywords:** annual and semi-annual oscillations; Taylor–Couette; sea level pressure

## 1. Introduction

This paper is an attempt to obtain better constraints on the forcings of the trend and annual components of both global sea-level pressure and variations in the Earth's rotation and to test the hypothesis that there might be a causal link between them.

Lopes et al. [1] studied the singular spectral analysis (SSA) of the evolution of mean sea-level atmospheric pressure (SLP) since 1850. In addition to dominant trends, eleven

quasi-periodic components were identified, with (pseudo-) periods of $\sim$130, 90, 50, 22, 15, 4, 1.8, 1, 0.5, 0.33, and 0.25 years, corresponding to the Jose, Gleissberg, Hale, and Schwabe cycles, to the annual cycle, and its first three harmonics. These periods are already known to be characteristic of the space-time evolution of the Earth's rotation axis: the rotation pole (RP) also undergoes periodic motions longer than 1 year, at least up to the Gleissberg $\sim$90 years cycle [2–15]. They are encountered in solar physics [16–25] and terrestrial climate [26–35].

The rotation velocity is usually expressed as the length of day (lod); it contains periods of 1 year and shorter, but also (although they are weaker) longer periods from 11 years to 18.6 years [36–40]. Lopes et al. [15] showed that the secular variation of lod since 1846 [41] and RP have been carried by an oscillation whose period fits the Gleissberg cycle.

RP consists to first order of three SSA components that together amount to 85% of the total signal variance [12]: the drift [4], the free oscillation [2,3] (actually a double component with periods $\sim$433 and $\sim$434 days), and the forced annual oscillation [7]. Polar drift does not have a universally accepted explanation [5,9,42,43]. The two other oscillatory components (Chandler and annual) are obtained using the Liouville–Euler system, which describes the motion of a spherical rotating solid body (e.g., [7], chapter 3, system 3.2.9).

Polar drift was discussed in [13,15], and the free Chandler wobble was addressed in [14]. In the present paper, we focus on the third major component, the forced annual oscillation. This oscillation is often assumed to be forced by the Earth's fluid envelopes, hence also by climate variations (e.g., [7,44], chapter 7, [45,46]). The fact that the observed Chandler period (433–434 days) is not equal to the theoretical value of the Euler period (306 days) has been interpreted in terms of Earth elasticity. The annual period is forced by interactions with fluids. Variations in the Sun–Earth distance, hence of the corresponding gravitational forces, displace the fluid atmosphere and ocean; exchanges in angular moment affect RP and lod. The annual oscillation of the rotation pole RP (coordinates $m_1$ and $m_2$) would therefore be due to the presence of the fluid envelopes. If Earth were devoid of fluid envelopes, the annual oscillation should not exist, which would contradict both the theory and observations of motion of a Lagrange top [47]. Wilson et Haubrich [48] recall that Spitaler [49,50] demonstrated that a part of the forcing of the annual wobble was due to the migration of air masses on and off the Asian continent. The authors in [51] showed that the annual fluctuation in water storage on the continents was also important. Study [52] re-examined the sources of annual wobble excitation and concluded that the air mass effect accounted for much but not all of the annual wobble, and water storage did not explain the remainder. Reservations appear in chapter 7 of Lambeck 's book [7], Seasonal Variations, page 146, who writes that since Jeffreys, in 1916, the seasonal oscillation has been attributed to a geographical redistribution of mass associated with meteorological causes (e.g., atmospheric and oceanic motion, precipitation, vegetation, polar ice, etc.) but, although this conclusion is still valid today, it presents some important discrepancies when we compared observed and computed annual components. In summary, the interpretation of the forced annual component of polar motion is still hypothetical and fails to be validated by a numerical model.

In a previous analysis of global sea-level pressure (SLP), we found that trends since 1950 were very stable in time and space [1] and were organized in a dominant 3-fold symmetry about the rotation axis in the northern hemisphere (NH) and a 3 or 4-fold symmetry in the southern hemisphere (SH) [1]. These features could be interpreted as resulting from Taylor–Couette flow. In this paper, we return to the pressure data and focus on the annual oscillation.

## 2. The Pressure Data and Method of Analysis

The pressure data are maintained by the Met Office Hadley Centre (https://www.metoffice.gov.uk/hadobs/hadslp2/data/download.html, accessed on 10 October 2020) and can be accessed as maps of global pressure, every month from 1850 up to the present. The sampling is $5° \times 5°$; the data we use are labeled HadSLP2r. Allan and Ansell ([53])

built this series starting with ground observations whose number and location are shown in their figure 2. These are subjected to a quality check, corrected for various local effects, then homogenized with the empirical mode decomposition (**EMD**) filter.

As in [1], we have used the iterative singular spectrum analysis algorithm (**iSSA**) to extract the components, mainly the annual one, in each grid cell and as a function of time. For the **iSSA** method, see [54]; for properties of the Hankel and Toeplitz matrices, see [55]; and for the singular value decomposition (**SVD**) algorithm, see [56]. We will now summarize the four steps of the SSA method.

Let us consider a discrete (non zero) time series ($\mathcal{X}_N$) of length $N$ ($N > 2$):

$$\mathcal{X}_N = (x_1, \ldots, x_N). \tag{1}$$

*2.1. Step 1 (Embedding Step)*

The time series $\mathcal{X}_N$ is divided into $K$ segments of length $L$ in order to build a matrix **X** with dimension $K \times N$, where $K = N - L + 1$ will condition our decomposition. This is the first "tuning knob". Integrating **X** yields a Hankel matrix:

$$\mathbf{X} = \begin{pmatrix} x_1 & x_2 & x_3 \cdots & x_K \\ x_2 & x_3 & x_4 \cdots & x_{K+1} \\ x_3 & x_4 & x_5 \cdots & x_{K+2} \\ \vdots & \vdots & \vdots \ddots & \vdots \\ x_L & x_{L+1} & x_{L+2} \cdots & x_N \end{pmatrix} \tag{2}$$

Embedding, the first step in an SSA, consists of projecting the one-dimensional time series in a multidimensional space of series $\mathcal{X}_N$ such that vectors $X_i = (x_i, \ldots, x_{i+L-1})^t$ belong to $\mathcal{R}^L$, where $K = N - L + 1$. The parameter that controls the embedding is $L$, the size of the analyzing window, an integer between 2 and $N - 1$. The Hankel matrix has a number of symmetry properties: its transpose $\mathbf{X}^t$, called the trajectory matrix, has dimension $K$. Embedding is a compulsory step in the analysis of non-linear series. It consists formally in the empirical evaluation of all pairs of distances between two offset vectors, delayed (lagged) in order to calculate the correlation dimension of the series.

*2.2. Step 2 (Decomposition in Singular Values (SVD))*

The **SVD** [56] of non-zero trajectory matrix **X** (dimensions $L \times K$) takes the shape:

$$\mathbf{X} = \sum_{i=1}^{d} \sqrt{\lambda_i} U_i V_i^t \tag{3}$$

where the eigenvalues $\lambda_i (i = 1, \ldots, L)$ of matrix $\mathbf{S} = \mathbf{X}\mathbf{X}^T$ are arranged in order of decreasing amplitudes. Eigenvectors $U_i$ and $V_i$ are given by:

$$V_i = \mathbf{X}^T U_i / \sqrt{\lambda_i} \tag{4}$$

The $V_i$ form an orthonormal basis and are arranged in the same order as the $\lambda_i$. Let $\mathbf{X}_i$ be a part of matrix **X** such that:

$$\mathbf{X}_i = \sqrt{\lambda_i} U_i V_i^t. \tag{5}$$

Embedding matrix **X** can then be represented as a simple linear sum of elementary matrices $\mathbf{X}_i$. If all eigenvalues are equal to 1, then decomposition of **X** into a sum of unitary matrices is:

$$\mathbf{X} = \mathbf{X}_1 + \mathbf{X}_2 + \ldots + \mathbf{X}_d \tag{6}$$

with $d$ being the rank of X ($d = \text{rank } \mathbf{X} = max\{i | \lambda_i > 0\}$). **SVD** allows one to write **X** as a sum of $d$ unitary matrices, defined in a univocal way.

Let us now discuss the nature and the characteristics of the embedding matrix. Its rows and columns are sub-series of the original time series (or signal). The eigenvectors $U_i$ and $V_i$ have a time structure, and they can be considered as a representation of temporal data. Let **X** be a suite of $L$ lagged parts of ($\mathcal{X}$ and $X_1, \ldots, X_K$), the linear basis of its eigenvectors. If we let

$$Z_i = \sum_{i=1}^{d} \sqrt{\lambda_i} V_i \tag{7}$$

with $i = 1, \ldots, d$, then relation (5) can be written:

$$\mathbf{X} = \sum_{i=1}^{d} U_i Z_i^t \tag{8}$$

that is, for the $j$th elementary matrix:

$$X_j = \sum_{i=1}^{d} z_{ji} U_i \tag{9}$$

where $z_{ji}$ is a component of vector $Z_i$. This means that vector $Z_i$ is composed of the $i$th components of vector $X_j$. In the same way, if we let

$$Y_i = \sum_{i=1}^{d} \sqrt{\lambda_i} U_i \tag{10}$$

we obtain for the transposed trajectory matrix:

$$X_j^t = \sum_{i=1}^{d} U_i Y_i^t \tag{11}$$

that corresponds to a representation of the $K$ lagged vectors in the orthogonal basis ($V_1, \ldots, V_d$). One sees why SVD is a very good choice for the analysis of the embedding matrix, as it provides two different geometrical descriptions.

### 2.3. Step 3 (Reconstruction)

As we have seen, $\mathbf{X}_i$ matrices are unit matrices, and one can "re-group" these matrices into a physically homogeneous quantity. This is the second "tuning knob" of **SSA**. In order to regroup the unit matrices, one divides the set of indices $i\{1, \ldots, d\}$ into $m$ disjoint subsets of indices $\{I_1, \ldots, I_m\}$.

Let $I$ be the grouping of $p$ indices of $I = \{i_1, i_2, \ldots, i_p\}$; because (6) is linear, then the resulting matrix $\mathbf{X}_I$ that regroups indices I can be written:

$$\mathbf{X}_I = \mathbf{X}_{I1} + \mathbf{X}_{I2} + \ldots + \mathbf{X}_{Im} \tag{12}$$

This step is called regrouping the eigen-triplets ($\lambda$, $U$, and $V$). In the limit case $m = d$, (12) becomes exactly (6), and we find again the unit matrices.

Next, how can one associate pairs of eigen-triplets? This means separating the additive components of a time series. One must first consider the concept of separability.

Let $\mathcal{X}$ be the sum of two time series $\mathcal{X}^{(1)}$ and $\mathcal{X}^{(2)}$ such that $x_i = x_i^{(1)} + x_i^{(2)}$ for any $i \in [1, N]$. Let $L$ be the analyzing window (with fixed length), $X$, $X^{(1)}$, and $X^{(2)}$ the embedding matrices of series $\mathcal{X}$, $\mathcal{X}^{(1)}$, and $\mathcal{X}^{(2)}$. These two sub-series are separable (even weakly) in Equation (6) if there is a collection of indices $\mathcal{I} \subset \{1, \ldots, d\}$ such that $\mathbf{X}^{(1)} = \sum_{i \in \mathcal{I}} \mathbf{X}_i$, respectively, if there is a collection of indices such that $\mathbf{X}^{(1)} = \sum_{i \notin \mathcal{I}} \mathbf{X}_i$.

In the case when separability does exist, the contribution of $\mathbf{X}^{(1)}$, for instance, corresponds to the ratio of associated eigenvalues ($\sum_{i \in \mathcal{I}} \lambda_i$) to total eigenvalues ($\sum_{i=1}^{d} \lambda_i$).

Therefore, regrouping SVD components can be summarized by the decomposition into several elementary matrices, whose structure must be as close as possible to a Hankel matrix of the initial trajectory matrix (this is true on paper only; in reality things are much more difficult).

*2.4. Step 4 (The Diagonal Mean, Aka the Hankelization Step)*

The next, final step consists in going back to data space, that is, to calculate time series with length N associated with sub-matrices $\mathbf{X}_I$. Let $\mathbf{Y}$ be a matrix with dimension $L * K$ and for each element $y_{ij}$ we have $1 \leqslant i \leqslant L$ and $1 \leqslant j \leqslant K$. Let $L^*$ be the minimum and $K^*$ be the maximum. One always has $N = L + K - 1$. Finally, let $y_{ij}^* = y_{ij}$ if $L < K$ and $y_{ij}^* = y_{ji}$ otherwise. The diagonal average applied to $k$th index of time series $y$ associated with matrix $\mathbf{Y}$ gives:

$$
y_k = \begin{cases} \frac{1}{k} \sum\limits_{m=1}^{k} y_{m,k-m+1}^* & 1 \leqslant k \leqslant L^* \\ \frac{1}{L^*} \sum\limits_{m=1}^{L^*} y_{m,k-m+1}^* & L^* \leqslant k \leqslant K^* \\ \frac{1}{N-K+1} \sum\limits_{m=k-K*+1}^{N-K*+1} y_{m,k-m+1}^* & K^* \leqslant k \leqslant N^* \end{cases} \tag{13}
$$

The relation (13) corresponds to the mean of the element on the anti-diagonal $i + j = k + 1$ of the matrix. For $k = 1$, $y_1 = y_{1,1}$. For $k = 2$, $y_2 = (y_{1,2} + y_{2,1})/2$, etc.

As noted above, step 3 is the most difficult part. We have chosen one approach among many others: iterative SSA (**iSSA**). Because relation (6) is linear, we can iterate the decomposition. We start with a small value of $L$ (we are looking for the longest period) that we increase until getting a quasi-Hankel matrix (step 1 and 2). We then extract the corresponding lowest frequency component that it subtracted from the original signal. We increase again the value of $L$ to find the next component (shortest period). The algorithm stops when no pseudo-cycle can be detected or extracted. In this way, we scan the series from low to high frequencies.

## 3. The SSA Pressure Components

We obtain the following components:

The trend (from ~1009.15 hPa in 1850 to ~1008.35 hPa at present) is the first and largest iSSA component. It represents more than 70% of the total variance of the original series. The sequence of the quasi-periodic components is, in decreasing order of periods (amplitudes are in hectoPascal, hPa):

- ~130 years (~0.7 hPa). Compatible with the Jose [17] cycle,
- ~90 years (~21 hPa). We recognize the Gleissberg [16] cycle [21],
- ~50 years (~0.2 hPa),
- ~22 years (~0.50 hPa). We recognize the Hale cycle [57],
- ~15 years (~0.2 hPa). We may recognize an upper bound of the Schwabe cycle [58],
- ~4 years (~0.3 hPa),
- ~1.8 years (~0.3 hPa).

Then:

- 1 year ( ~93 hPa),
- 0.5 years (~65 hPa),
- 0.33 years (~44 hPa),
- 0.25 years (~21 hPa).

## 4. Annual and Semi-Annual SSA Pressure Components and Variations in Polar Motion: Time Analysis

Many studies (e.g., [7,51,59]) attempt to relate forced (annual) oscillations to polar motion RP ($m_1$, $m_2$) or length of day (lod) (data file EOPC01IAU2000, [60], maintained by

IERS, https://www.iers.org/IERS/EN/DataProducts/EarthOrientationData/eop.html, accessed on 12 October 2020). The data are shown and discussed in [14]. The annual iSSA components of $m_1$ and $m_2$ are displayed in Figure 1. Both undergo some amount of modulation. That modulation is significant for $m_1$, particularly between 1850 and 1930. For $m_2$, it is smaller and more regular. An $m_1$ vs. $m_2$ diagram (see Appendix A) shows the classical Lissajou elliptical shape of annual polar motion. Note that the date of 1930 is when the Chandler wobble undergoes a phase jump of $\pi$ (e.g., [61,62]). The lod data start only in 1962 (not shown, see [15]).

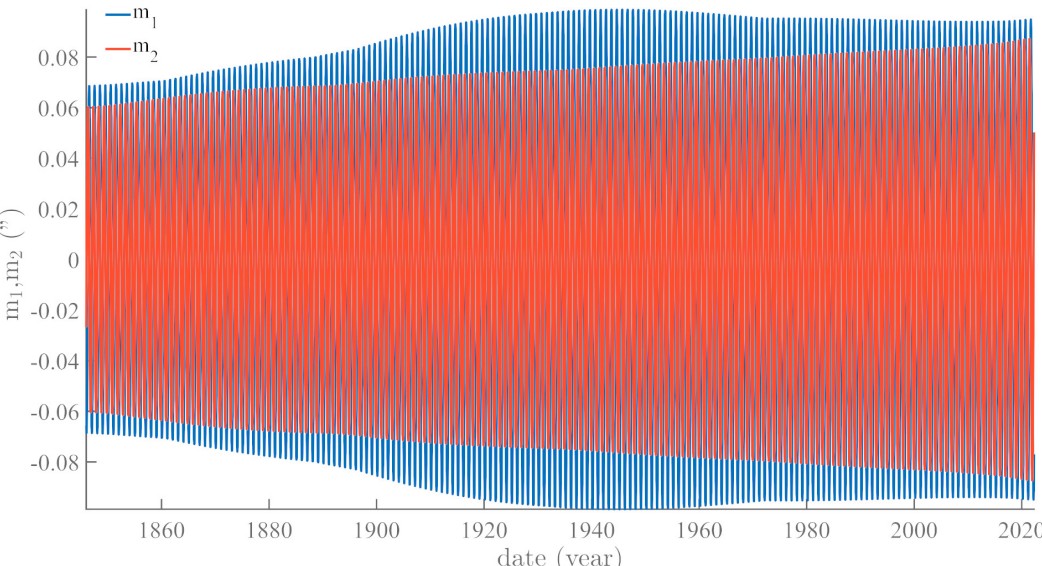

**Figure 1.** The iSSA annual component of rotation pole coordinates $m_1$ and $m_2$ from 1850 to the present.

The annual and semi-annual iSSA components of lod are modulated (Figure 2a,b). The annual component grows slightly until 1990, flattens until 2013, and then starts growing again. The semi-annual component decreases regularly over the sixty years of available data.

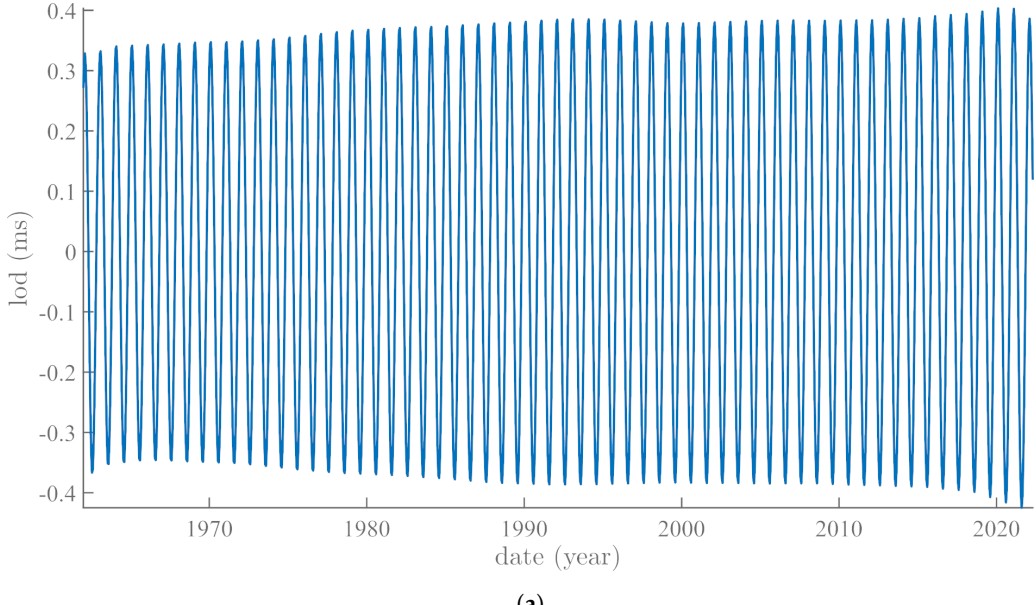

(a)

**Figure 2.** *Cont.*

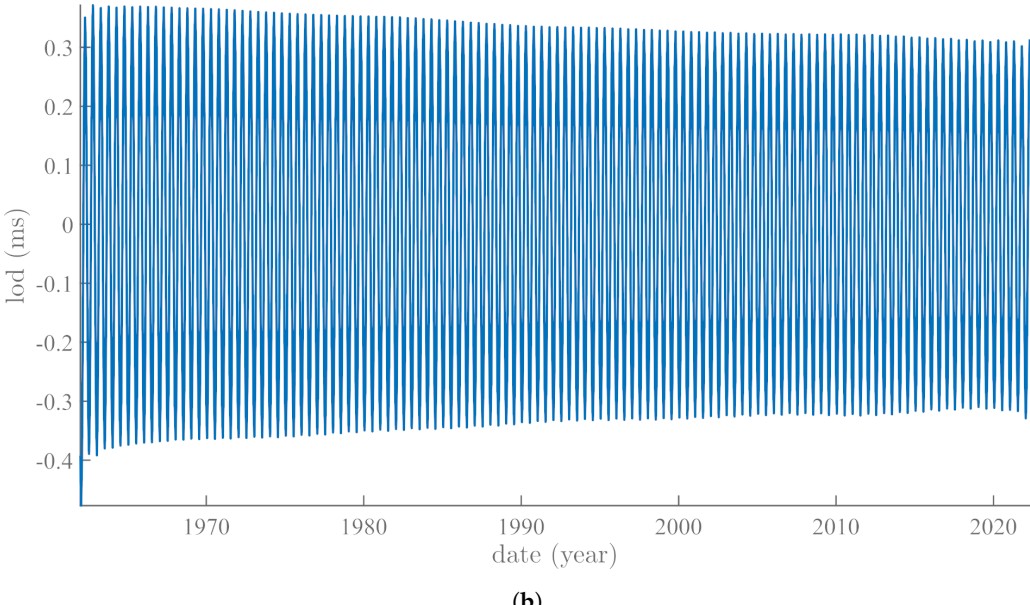

**(b)**

**Figure 2.** Annual and semi-annual components extracted from length of day: (**a**) iSSA annual component of length of day from 1850 to the present; (**b**) same as (**a**) but for the semi-annual component.

Figure 3 shows the annual iSSA component since 1850, extracted from the SLP pressure series. Its modulation looks very much like that of $m_2$, with a phase shift (Figure 1). We have checked that the phase shift is constant over the 70 years of data. Its mean value is $152.31 \pm 2.68$ days. We have already obtained (exactly) such a phase shift and note that it is (exactly) half of the Euler period of 306 days: a reason for this remains to be found.

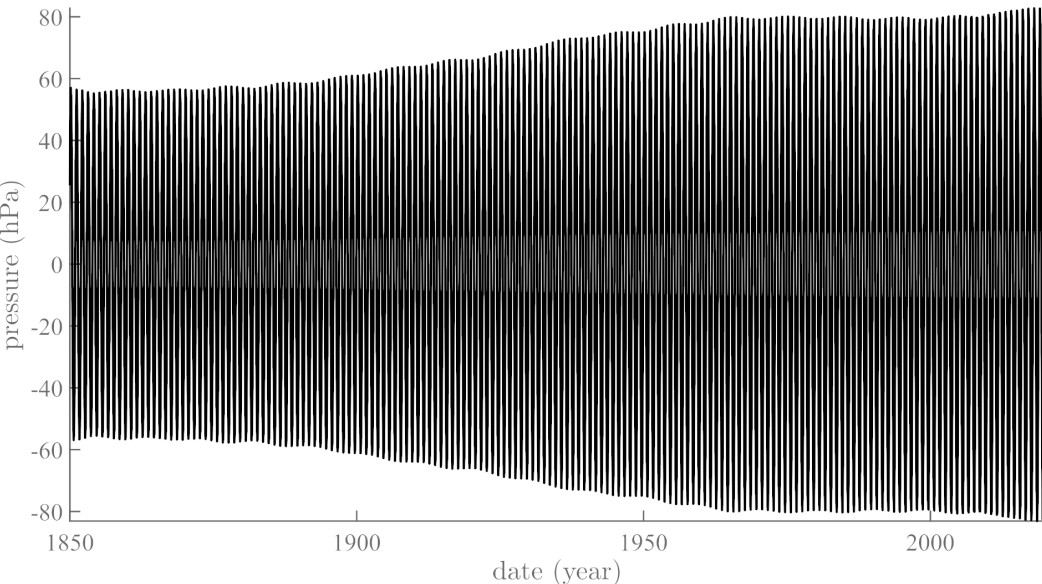

**Figure 3.** The iSSA annual component of sea level pressure SLP from 1850 to the present.

Using the Hilbert transform, we have determined the envelopes of the two annual components of polar motion and atmospheric pressure (Figure 4a). Polar motion clearly leads pressure by at least 40 years. In Figure 4b, we see that the trend of polar motion is close to the envelope of pressure, in general preceding it slightly. Given the mathematical properties of the Liouville–Euler set of equations, the potential direction of causality between the two phenomena is in the sense of polar motion leading atmospheric pressure.

If correct, this is an important result that confirms Laplace's statements in his 1799 Traité de Mécanique Céleste (volume 5, chapter 1, page 347) [63].

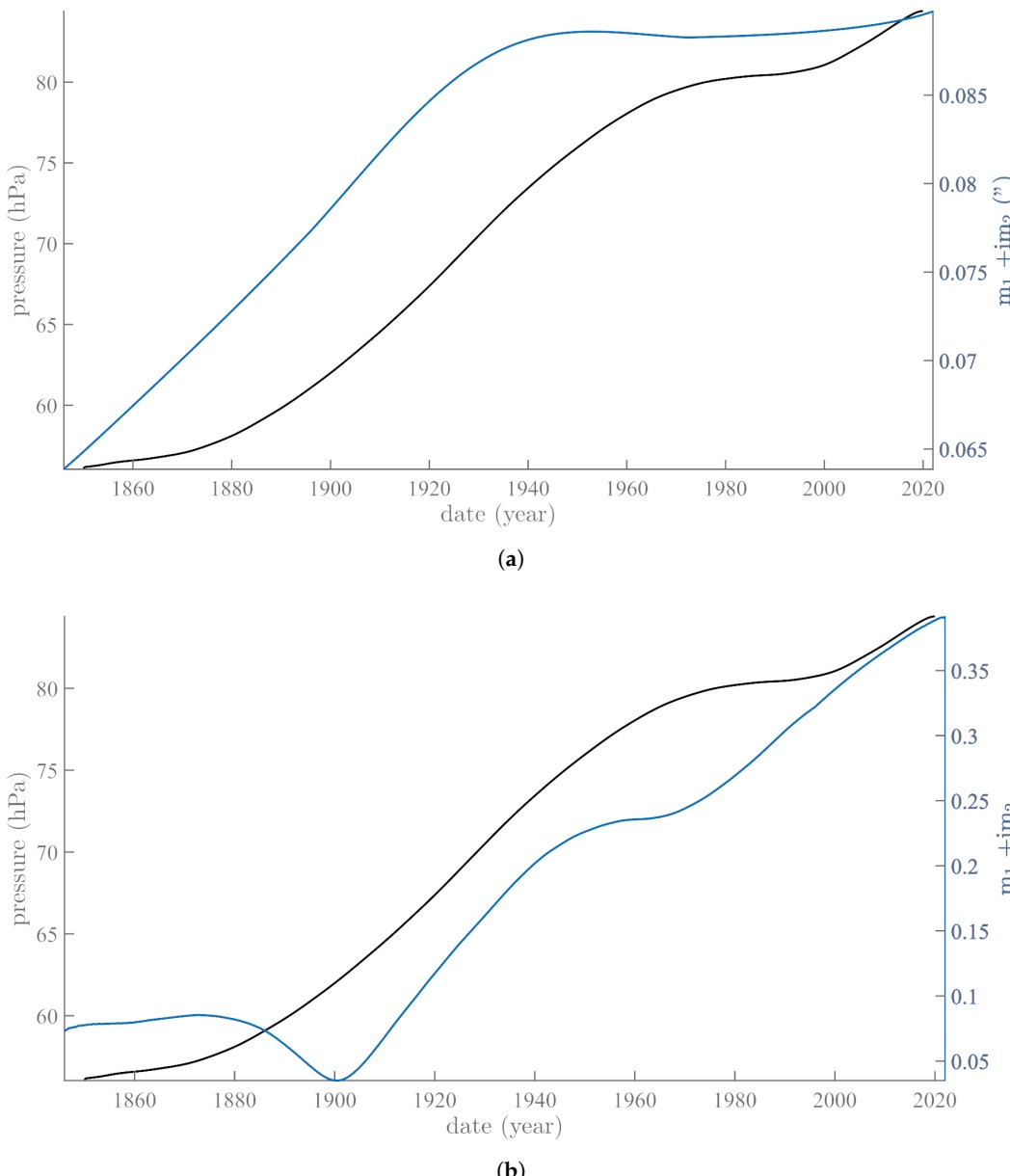

(**a**)

(**b**)

**Figure 4.** Annual envelopes and trends extracted from pole movement and **SLP**. (**a**) Envelopes of oscillations of iSSA annual components of polar motion *m* (blue curve, right scale) and global sea-level pressure **SLP** (black curve, left scale). (**b**) Trend of the pole movement (blue curve) and envelope of iSSA component of atmospheric pressure SLP (black curve).

Thus, after applying the Kepler's law of conservation of areas, Laplace concludes that atmospheric motions do not affect pole rotation.

## 5. The Annual and Semi-Annual SSA Pressure Components: Spatial Analysis

We show in Figure 5 polar maps of the mean amplitude of the annual component (oscillation) for each season since 1850, and in Figure 6a–d polar maps of the mean of the semi-annual component.

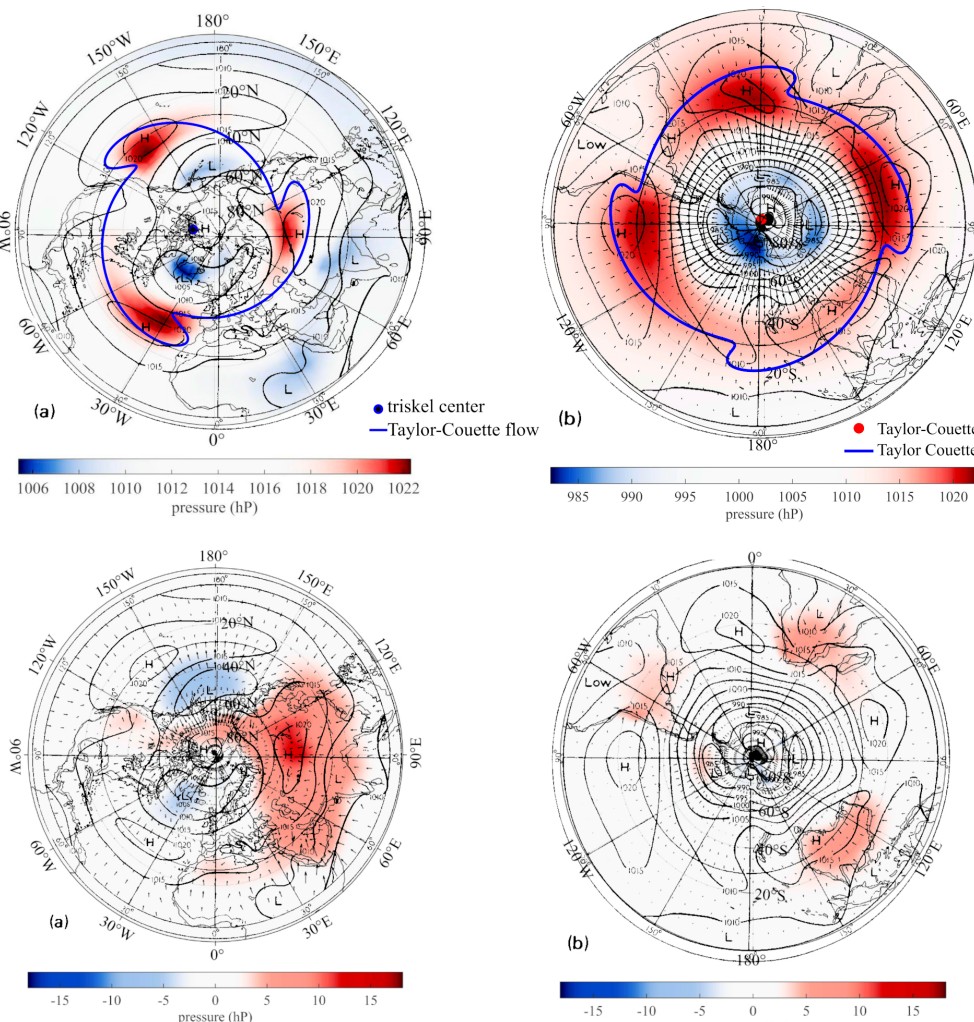

**Figure 5.** Overlay of the triskeles patterns (SSA trend, top) and SSA annual oscillation of SLP (bottom) from Lopes et al. (2022) with maps from Lamb (1972) page 157, Figures 4–13. Left, southern hemisphere; right, southern hemisphere. Lamb's maps are calculated for a time span of 40 years; our SSA results are for 170 years.

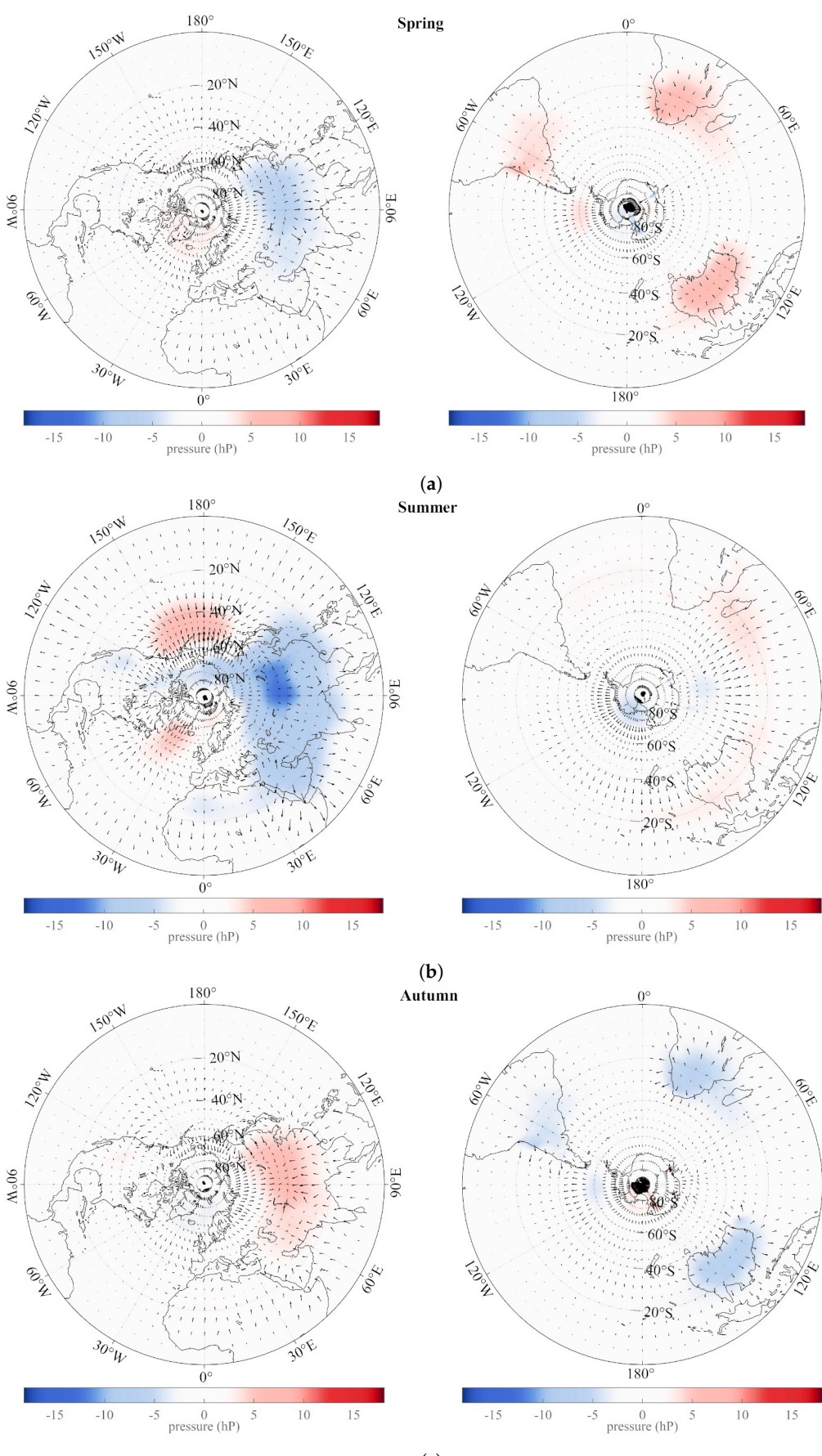

**Figure 6.** *Cont.*

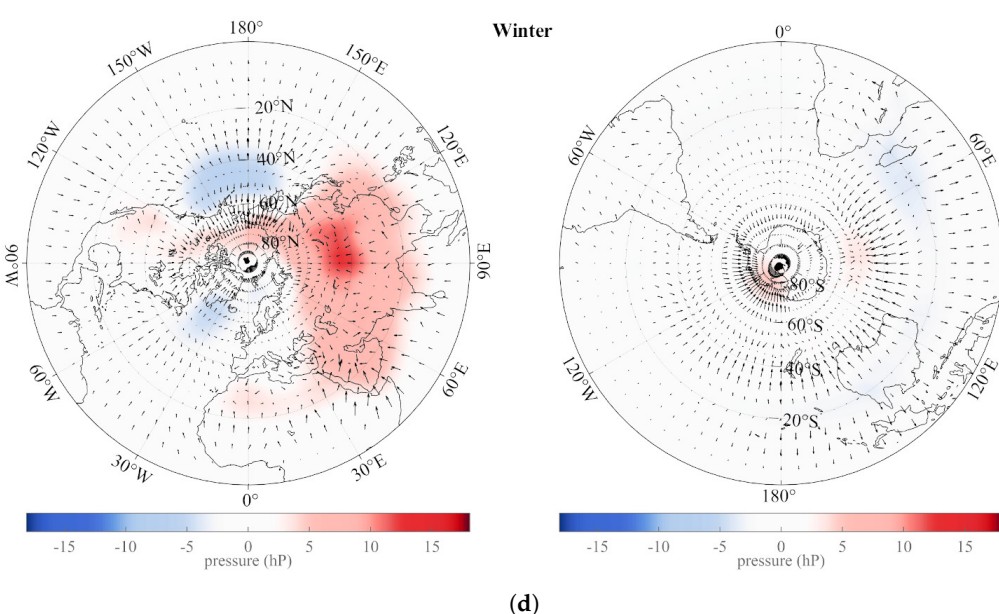

**Figure 6.** Polar stereographic projection of the northern hemisphere (left column) and southern hemisphere (right column) showing the mean (1850–2020) of the annual oscillation (**iSSA** component 2) for all (from top to bottom) springs, summers, autumns, and winters since 1850: (**a**) during springs; (**b**) during summers; (**c**) during autumns; (**d**) during winters.

In Figure 6a–d, in the spring SH, there is, in addition, a very regular pattern at the intersections of the 20° S to 50° S annulus and the southern tips of the three southern continents (Africa, Australia, and South America). There is a smaller dipole with its negative part centered almost perfectly on the South Pole and its positive part centered on 70° S, 105° W. In the fall, the pattern is the same but with a reversal in sign. In the summer, there is a positive band between 15 and 25° S latitude from Madagascar to New Caledonia (40°–170° E) and a small negative spot near the South Pole (winter is the same with a sign reversal). The positive annulus is actually also seen, much weaker, in the Atlantic and Pacific oceans. The NH is rather different in the spring, with a large negative area over Asia from the Red Sea to Japan and extending in latitude from 80° N to 15° N. This large "anomaly" (using geophysical language) extends negative arms towards Senegal and over the western USA. Over the year, the "anomaly" changes sign. Still, in the summer, there are two sizable positive anomalies over the northern Atlantic and Pacific Oceans between 40° N and 60° N. In the SH, the large mean annual pressure variations occur over the southern continents; in the NH, the Asian continental "anomaly" dominates, although two significant features occur over the northern parts of the oceans. Because the Asian "anomaly" and the two (opposite sign) anomalies in the North Atlantic and Pacific are separated by almost 180° in longitude, their maxima being always in phase, their contributions to the seasonal excitation function tend to cancel [64].

The semi-annual component (see Figure 7) has stronger patterns of symmetry. The SH features a very stable 3-fold symmetry (triskeles); three large anomalies form an equilateral triangle centered on 70° E, 30° W, and 210° W, between latitudes 40° S and 60° S, a large anomaly of opposite sign is centered on the South Pole (the maximum is actually offset from the SP by 20° E towards the 120° E meridian), and an anomaly with the same sign as that over the pole, but located over southern Africa. The NH has weaker anomalies forming an irregular triangle with apices near the North Pacific (40° N, 165° E), Baffin Sea (40° N, 50° W), and Iran (30° N, 60° W).

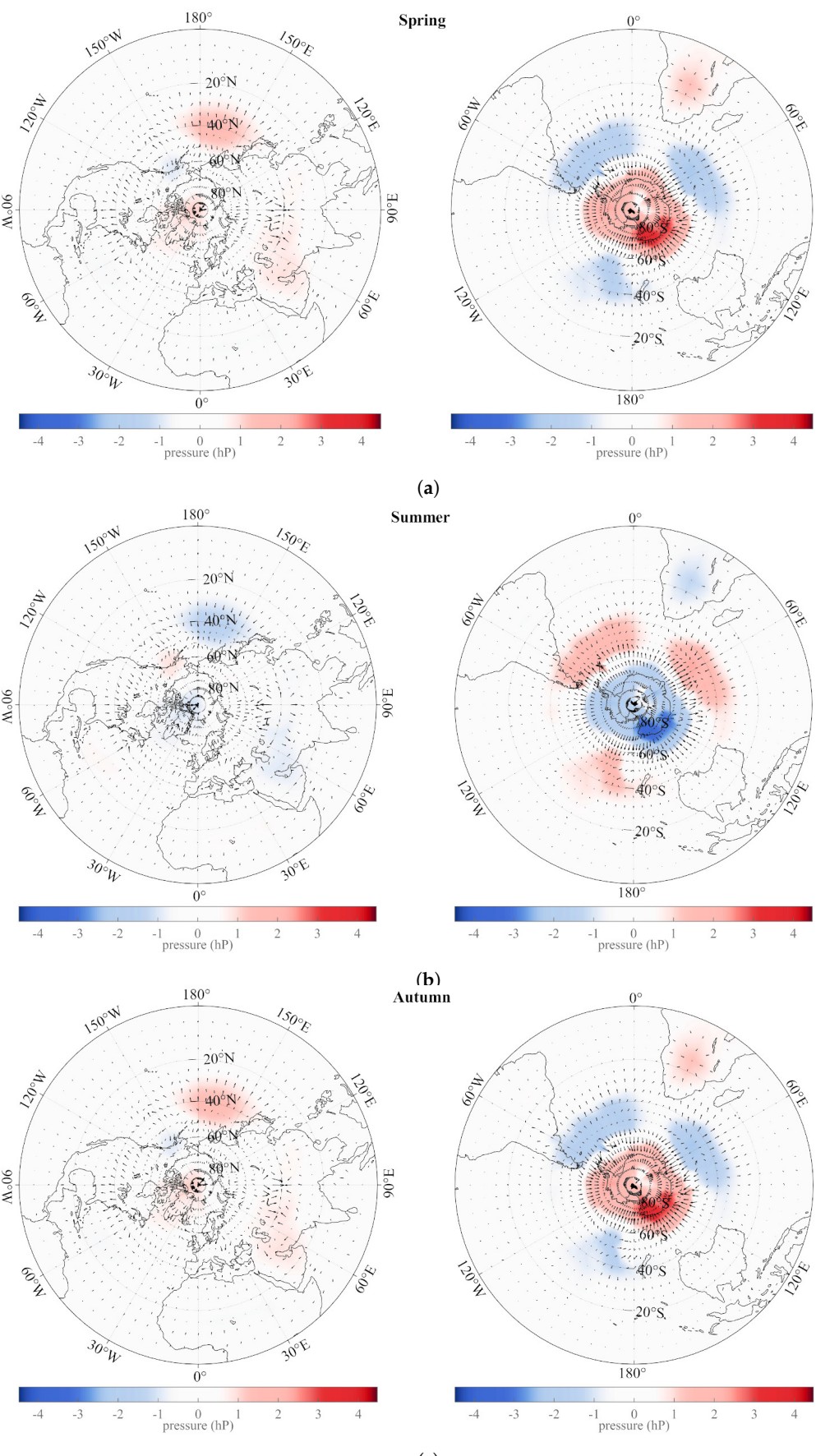

**Figure 7.** *Cont.*

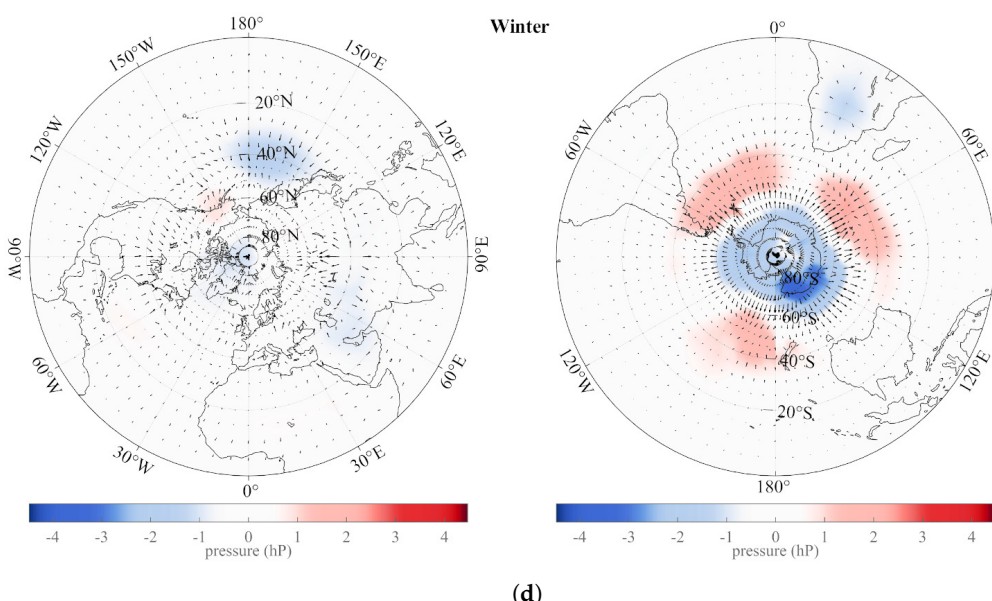

**(d)**

**Figure 7.** Same as Figure 5 but for the semi-annual oscillation (SSA component 3): (**a**) during springs; (**b**) during summers; (**c**) during autumns; (**d**) during winters.

The bottom two maps in Figure 8b–c are the spring maps for the annual and semi-annual components.

It is interesting to compare the maps of Figure 8 with the maps of mean pressure gradients shown in figure 2 of Lopes et al. [1]. The zonal structure appears clearly, with sharp changes in the sign of anomalies at continent/ocean edges. The zonal structure is reminiscent of the bands seen in the atmosphere of the giant planets; the continent/ocean boundaries interrupt the zonal belts, leading to a segmented structure.

We can compare the maps of Figures 5–8 for the annual, semi-annual, and trend components with corresponding figures from the reference treatise of [64]. Recall that, taken together, these three components suffice to capture the total signal variance. Figures 4–12 and Figures 4–13 ([64], pages 156–157 and reproduced here Figure 5) show Lamb's representation of the average mean sea level pressure over (respectively) the northern and southern hemispheres in the 1950s.

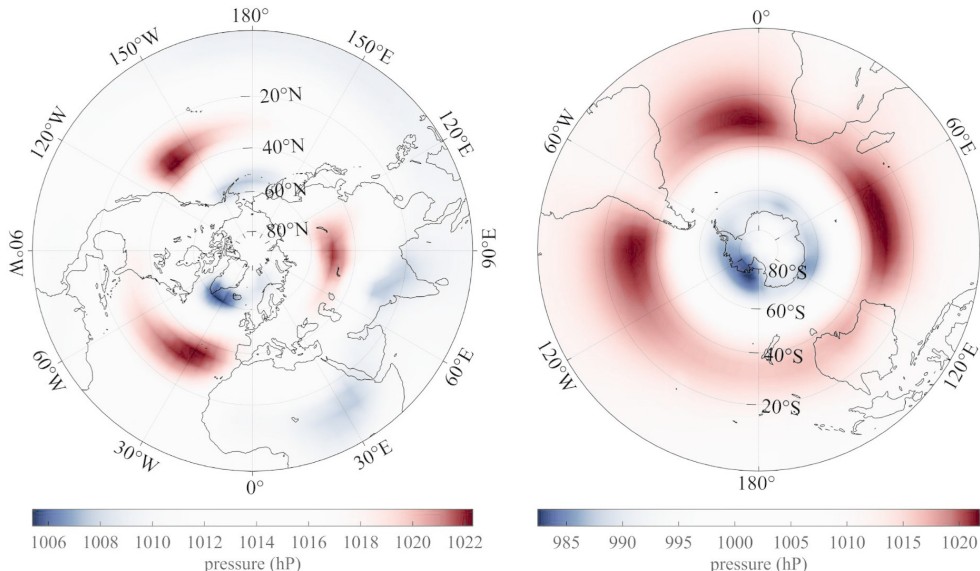

**Figure 8.** *Cont.*

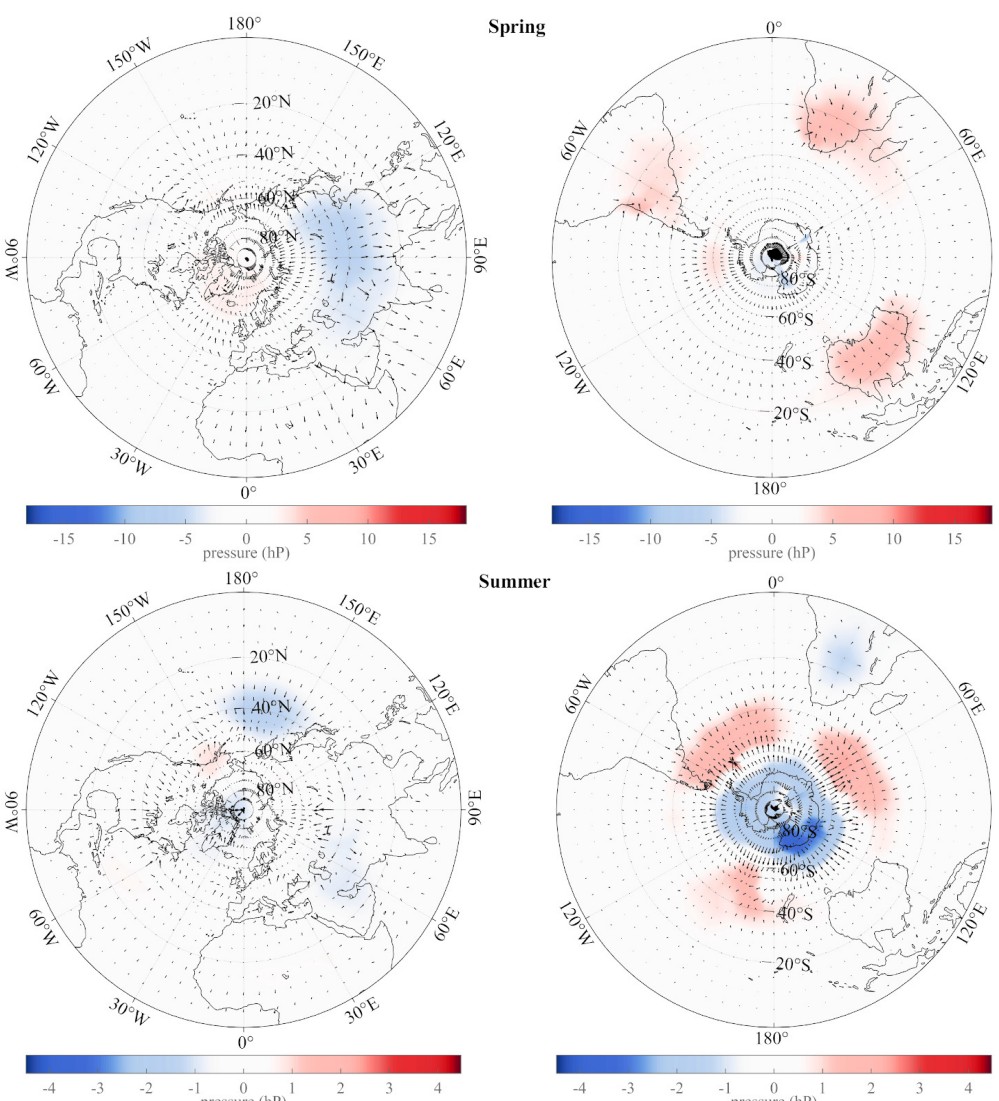

**Figure 8.** Polar views of northern hemisphere (left column) and southern hemisphere (right column) showing the trend (iSSA component 1 from Lopes et al. [1]), and maps of iSSA components 2 (annual) and 3 (semi-annual) for all springs since 1850.

Figure 5 shows an overlay of the SSA triskeles patterns (trends) and annual oscillation of SLP from Lopes et al. ([1]) on top of maps from Lamb (1972, page 157, Figures 4–13). The left column is for the northern hemisphere, the right one for the southern hemisphere. Lamb's maps are calculated for a time span of 40 years; our SSA results are for 170 years. There is very good agreement for the northern hemisphere: the January map from Lamb and our winter map show the dominant "dipole" with the large high pressure (HP) over Asia and the low pressure (LP) over the northern Pacific. The April and spring maps do not agree as well. The July map from Lamb and our summer map both show a large LP over Asia and two HPs over the northern Pacific and northern Atlantic. The October map from Lamb and our autumn map show a weaker HP over Asia. For the southern hemisphere, agreement between the two sets of maps is significantly lower; the winter vs. January maps both feature a HP centered on Antarctica and a LP extending from Madagascar to Australia, and the reverse in summer/July. Our spring (respectively, autumn) maps feature strong HPs (respectively, LPs) on the southern tips of the southern continents. This pattern is not as sharp in Lamb's (1972) maps.

Lamb's Figures 3–17 ([64] and Figure 5) represent the annual mean distribution of atmospheric pressure at sea level for the time span 1900–1940 for the northern hemisphere

and 1900–1950s for the southern hemisphere. This can be compared to our maps of the trend ([1], Figure 6 top row and Figure 7). The agreement is quite good: the 3-fold symmetry of the northern hemisphere and the 4-fold symmetry of the southern hemisphere are even clearer in the [1] maps using the iSSA method to extract the trend (i.e., the main component) from the data. As concluded by [64], the effects of geography modify the circumpolar circulation much more in the northern than in the southern hemisphere; "The broad zonal character of the long-period mean circulation is as permanent as the circumpolar arrangement of the heating and cooling zones and the circumpolar vortex aloft". The patterns we obtain in 2020 and those Lamb (1972) [64] obtained 50 years before are so similar that one can conclude the multidecadal stability of the atmospheric circulation. Our maps are based on 170 vs. 40 years of data, which lends credence to the sharper and simpler features of our maps. It appears to us that such maps from the southern hemisphere are particularly interesting, and so is the triskeles in Figure 5 (top left). Specialists may wish to analyze these maps in more detail; they can be provided on request.

## 6. Discussion

### 6.1. Symmetries and Forcings

We pointed out in a preliminary paper on symmetries of global sea level pressure (Lopes et al. (2022b) [1]) that the pattern of Figure 8 was remarkably stable over the 150 years of the data and that the observed geometry could be modeled as Taylor–Couette flow of mode 3 (NH) or 4 (SH). The remarkable regularity and order three symmetry of the NH triskeles occurs despite the lack of cylindrical symmetry of the northern continents. The stronger intensity and larger size of features in the SH is linked to the presence of the annular currents. Following Kepler and Laplace (see above), it appears that the molecules in the atmosphere cannot perturb polar axis motion. The atmosphere behaves as a rotating cylinder undergoing stationary flow. The top of the cylinder is a free surface; the bottom is the surface of the solid crust and/or fluid ocean. This topography interacts strongly with the atmosphere.

### 6.2. Sun–Earth Distance and Phases

The seasonal change in sign agrees with changes in the Sun–Earth distance. The present values of the ellipticity of the Earth's orbit and the obliquity of its rotation axis lead to changes of orbital velocity with summer in the southern hemisphere at perihelion and summer in the northern hemisphere at aphelion. As discussed in [14,15], forces acting on the polar axis do so through a torque ($m_1$, $m_2$), obeying the Liouville-Euler system of equations (from Laplace (1799) [63]). The influence of these torques is in quadrature with the length of day (lod). Changes in orbital velocity of our planet lead to a variation in the solar torque, which acts against the sense of rotation of the planet. At perihelion, orbital velocity is 30.29 km/s and at aphelion 29.29 km/s. The torque associated with the Sun changes as this velocity changes. Thus, annual variations in lod should be in opposition to changes in the Sun–Earth distance, and they are (Figure 9a, bottom). In the northern latitudes, both the Earth's rotation and the winds go from west to east, the Earth's rotation slows in summer and accelerates in winter, hence there are positive pressure anomalies in winter, negative in summer. In other words, the pressure anomalies are due to the variation in rotation velocity (tracked through lod changes and winds). The phase difference between the annual components of lod and SLP is constant at 32 ± 1.2 days (Figure 9b). Relative variations of amplitudes should be similar, and indeed they are (Figure 9b).

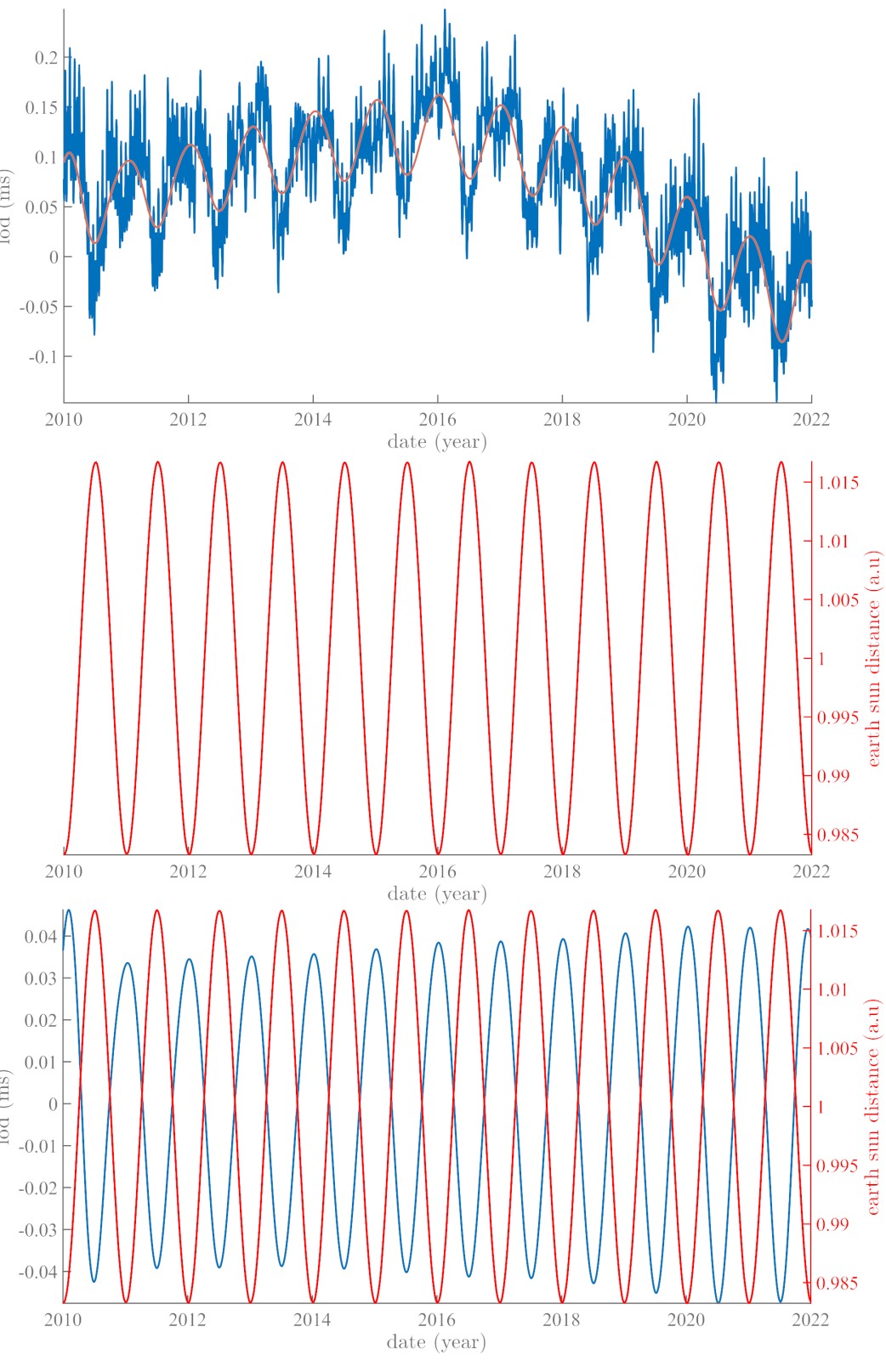

(**a**)

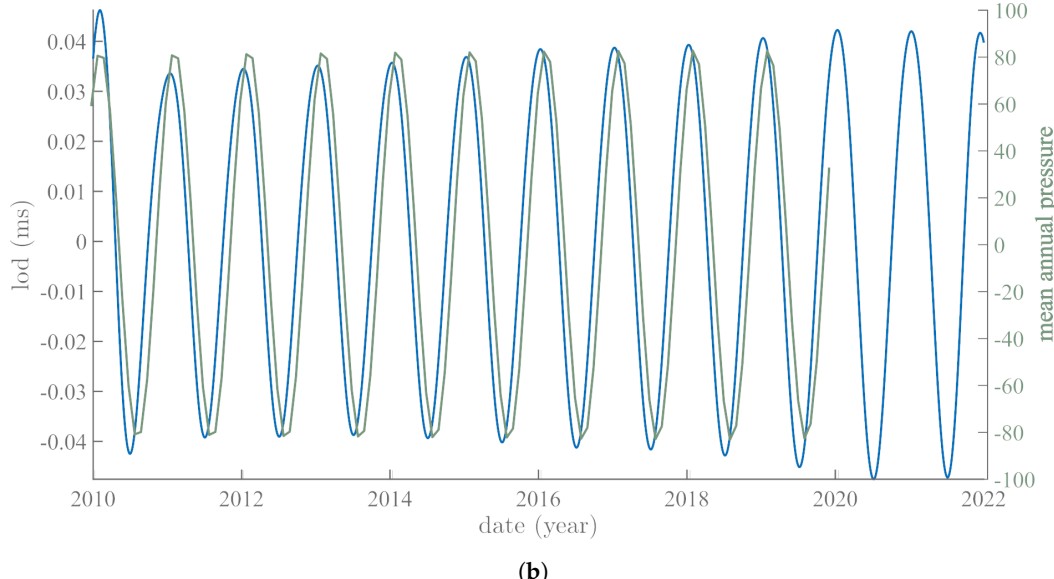

(**b**)

**Figure 9.** Astronomical and geophysical annual oscillation components: (**a**) (top) lod data (blue) and their annual component (red); (middle) variation in the Sun–Earth distance (red); (bottom) the two curves above superimposed, showing phase opposition; (**b**) in green, the annual SSA component 2 of global sea level pressure, and in blue, the annual component of lod. The phase difference is constant at $32 \pm 1.2$ days.

## 7. Sketch of a Mechanism

Because more than 70% of the total signal variance is captured by the trend of the atmospheric pressure, which varies by only 1 per mil since 1850, one can resort to the theory of turbulent flow in the case of infinitesimal perturbations (cf. [65]). The non-stationary $v_1(r, t)$ perturbation of the stationary solution $v_0(r)$ verifies the Navier–Stokes equation for an incompressible fluid:

$$\frac{\partial v_1}{\partial t} + (v_0 \nabla) v_1 + (v_1 \nabla) v_0 = -\frac{p_1}{\rho} + v.\mathrm{grad} v_1, \quad \mathrm{div} v_1 = 0. \tag{14}$$

The general solution can be written as the sum of particular solutions where the dependence of $v_1$ on time is of the form $e^{-i\omega t}$. There is still no theoretical basis for a mathematical solution of the stability of flow about finite dimension bodies plunged in a rotating fluid (e.g., [65–67]). Only experiments show that flow stability depends on the relative value of the Reynolds number ($R$), each type of flow having its critical Reynolds number ($R_c$), beyond which instability sets in (cf. [68]). It is still difficult to solve turbulent flows other than in the case of concentric cylinders (e.g., [69–75]). Fortunately, this is useful in the case of atmospheric and oceanic flows on the spherical Earth, as is suggested (among other evidence) by the annular structures shown in Figures 5–8.

Ref. [76] tackled the study of stability of a fluid comprised between two concentric rotating cylinders at high $R$. It was generalized for any value of $R$ by [77] that showed that particular independent solutions were given by:

$$v_1(r, \varphi, z) = f(r) * e^{i(n\varphi + kz - \omega t)} \tag{15}$$

with $n$ being an integer ($n$=0 for axial symmetry), $k$ the real wavelength of the instability, and $\omega$ an acceptable frequency solving the equations in plane $z$ = constant with $v_1$ = 0 on cylinders with radii $r = R_1$ and $r = R_2$ (see also the paper on triskeles by [15]). The Reynolds number is evaluated by $\Omega_1 R_1^2 / v$ (or $\Omega_2 R_2^2 / v$). For fixed $n$ and $k$, the $\omega$ become a discrete set of $\omega_n(k)$ values. In the case of the Earth, $v_1 = 0$ at the solid surface and at the atmosphere free surface, and the only observed solutions are 24 h and 1 year. We therefore expect to find harmonics of these two values. The detailed study for two

concentric cylinders can be found in [66]. The $v_1$ flow is stationary and consists of toric vortices (or Taylor vortices) regularly spaced along the cylinder generatrices. If the series of periods $\omega_n(k)$ linked to the annual oscillation is 1, 1/2, 1/3, 1/4 years (see Section 3), then one expects to find experimentally perturbations of order $n = 3$ or 4, as was already the case for trends ([1]; Figure 8a, top). We find in the present study that indeed such is the case for the annual (Figure 8b, middle) and semi-annual (Figure 8c, bottom) oscillations.

## 8. Conclusions

In this paper, we focus on the sources of the trend and annual components of both global sea-level pressure (SLP) and variations in the Earth's rotation (RP–coordinates of the rotation pole, and lod—length of day) and test the hypothesis that there might be a causal link between them. In a previous study [1], it was shown, using singular spectrum analysis (SSA), that the mean SLP contains, in addition to a weak trend, a dozen quasi-periodic or periodic components. These periods are characteristic of the space-time evolution of the Earth's rotation and are found in many characteristic features of solar and terrestrial physics. Polar drift and the free Chandler wobble have been studied in earlier papers ([13–15]). With the use of iterative SSA (iSSA), we have extracted the components of SLP in each grid cell and as a function of time since 1850. The trend averages 1009 hPa and varies by only 0.7 hPa over the 170 year period with available data. The amplitudes of the annual component and its harmonics decrease from 93 hPa for the annual to 21 hPa for the third harmonic. In contrast, the components with pseudo-periods longer than a year range between 0.2 and 0.5 hPa. The trend is as large as the annual component (21 hPa) and could be part of the 90 years Gleissberg cycle.

We have further analyzed the components of polar motion RP (coordinates $m_1$ and $m_2$) and length of day (lod). The annual components of RP and SLP have a phase difference of $152 \pm 2$ days, which is constant over 70 years; these components are modulated in the same way, growing in amplitude between ~1870 and ~1960. The phase difference happens to be half of the Euler period of 306 days, for reasons we do not yet know. There is a phase lag of ~40 years between RP and SLP. With the Euler–Liouville system of equations [14,63] in mind, we propose that there may be a causal link between the two.

We have mapped the first three iSSA components of global SLP that account for more than 85% of the total data variance. The trend pattern is stable, and the observed geometry can be modeled as Taylor–Couette flow of mode 3 (NH) or 4 (SH). The stronger intensity and larger size of features in the SH is linked to the presence of the annular mode (SAM). The annual component is characterized by a large negative anomaly extending over all of Eurasia in the NH summer (and the opposite in the NH winter) and three large positive anomalies over Australia and the southern tips of South America and South Africa in the SH spring (and the opposite in the SH autumn). The semi-annual component is characterized by three positive anomalies extending over Iran and the surrounding central Asia, the northwest Atlantic, and the northern Pacific in the NH spring and autumn (and the opposite in the NH summer and winter) and in the SH spring and autumn by a strong stable pattern consisting of three large negative anomalies forming a triskeles within the 40°–60° annulus formed by the southern oceans. A large positive anomaly centered over Antarctica, with its maximum actually displaced toward Australia, and a smaller one centered over Southern Africa complement the pattern. The pattern is opposite in the NH summer and winter.

The main features that appear in the maps underline the importance of the geographical patterns of continent–ocean boundaries and of the occurrence of (zonal) annular features (particularly at latitudes south of 20° S).

To the first order, large scale atmospheric motions can be modeled as rotating cylinders, stable in time and space, in the shape of 3 or 4 mode triskeles. A component of flow forced by the Sun–Earth distance modifies slightly polar rotation leading to seasonality of pressure anomalies. Because lod is physically linked to the solid Earth, geographical regions with strong topography are most affected by these variations.

The present values of the ellipticity of the Earth's orbit and the obliquity of its rotation axis lead to changes in orbital velocity. Forces acting on the polar axis do so through the torque ($m_1$, $m_2$). The torque changes as orbital velocity changes. The influence of this torque is in quadrature with the length of day (lod). Annual variations in lod are in opposition to changes in the Sun–Earth distance (Figure 9a, bottom). The phase difference between the annual components of lod and SLP is constant at $32 \pm 1.2$ days. Relative variations of amplitudes should be similar, and indeed they are (Figure 9b).

We propose that the pressure and rotational variations in the annual to multi-decadal range evidenced in increasing number and detail are, in significant part, driven by commensurabilities of the giant Jovian planets forcing variations on the Sun and in some cases directly on the inclination of the Earth's rotation axis and velocity. These variations, in turn, excite variations in a number of Earth parts (atmospheric pressure, sea-level, sea ice, and probably many more). They can be considered as a form of high-frequency Milankovitch "cycles". We intend to develop this point in a forthcoming paper.

**Author Contributions:** V.C., J.-L.L.M., F.L. and D.G. contributed to conceptualization, formal analysis, interpretation, and writing. All authors have read and agreed to the published version of the manuscript.

**Funding:** This research was supported by the Université de Paris, IPGP and the LGL-TPE de Lyon.

**Data Availability Statement:** The used data are freely available at the following address: Met Office Hadley Centre : https://www.metoffice.gov.uk/hadobs/hadslp2/data/download.html (accessed on 10 October 2020). IERS: https://www.iers.org/IERS/EN/DataProducts/EarthOrientationData/eop.html (accessed on 12 October 2020).

**Acknowledgments:** We thank the anonymous reviewers for their helpful comments on the manuscript.

**Conflicts of Interest:** The authors declare that they have no known competing financial interests or personal relationships that could have appeared to influence the work reported in this paper.

## Appendix A

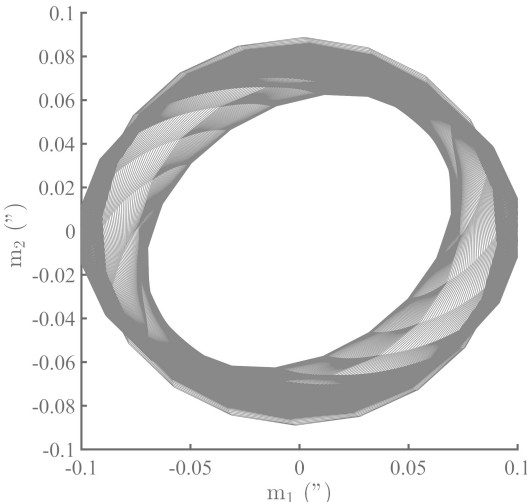

(**a**)

**Figure A1.** *Cont.*

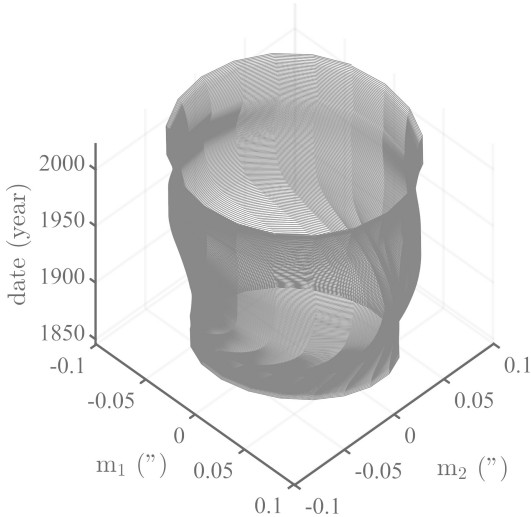

(**b**)

**Figure A1.** Spatio temporal evolution of the annual component of the rotation pole: (**a**) Lissajou pattern of the $m_1$ vs. $m_2$ coordinates of the iSSA annual component of the rotation pole; (**b**) same as in (**a**), with the added dimension of time (1850–2020).

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
