# Peer review of "On the Nature and Origin of Atmospheric Annual and Semi-Annual Oscillations"

_atmosphere, doi:10.3390/atmos13111907_

Round 1

Reviewer 1 Report

Please see my comments attached.

Author Response

Please find our answers in the uploaded pdf

Author Response

(The authors gave the same response as above.)

Round 2

Reviewer 1 Report

Please see my comments in the attachment.

Author Response

(The authors gave the same response as above.)

Round 3

Reviewer 1 Report

I feel that some reviewer's comments are very odd, but I do not have time to keep doing this and asking the authors to acknowledge a broad of previous climate community literature (studies) that have discussed/found that the origins/nature of annual and semi-annular oscillation in the atmosphere may also arise from the internal variability associated with eddy-mean flow interaction and other external forcings. Although in the end, the authors have acknowledged that some patterns they found are associated with NAM or SAM (see the revised version of their manuscript), none of the previous works have been cited.  Also, I just want to clarify that the work of Dommenget and Latif (2002) did not imply that you cannot trust the results of EOF or VARIMAX analysis, rather than cautions need to be paid when trying to interpret the statistically derived EOF modes and their significance. So, the periodic oscillation associated with a particular leading EOF mode can exist.   

I feel also rather ambivalent about this manuscript. It's interesting that the authors relate the annual and semi-annual oscillations of SLP to variation in Earth rotation, but other than this pervasiveness the findings are not new.

Nonetheless, considering the efforts they have made, I will give a chance for this present work to be published and lets the community assess it.

Best regards,